# Analysis of Gut Bacterial and Fungal Microbiota in Children with Autism Spectrum Disorder and Their Non-Autistic Siblings

**DOI:** 10.3390/nu16173004

**Published:** 2024-09-05

**Authors:** Mauricio Retuerto, Hilmi Al-Shakhshir, Janet Herrada, Thomas S. McCormick, Mahmoud A. Ghannoum

**Affiliations:** 1Department of Dermatology, Case Western Reserve University, Cleveland, OH 44106, USA; mar153@case.edu (M.R.); hilmi.al-shakhshir@emory.edu (H.A.-S.); jlh269@case.edu (J.H.); tsm4@case.edu (T.S.M.); 2Center for Medical Mycology, University Hospitals Cleveland Medical Center, Cleveland, OH 44106, USA

**Keywords:** autism spectrum disorder (ASD), microbiome, probiotic, regression analysis

## Abstract

Autism Spectrum Disorder (ASD) is a multifactorial disorder involving genetic and environmental factors leading to pathophysiologic symptoms and comorbidities including neurodevelopmental disorders, anxiety, immune dysregulation, and gastrointestinal (GI) abnormalities. Abnormal intestinal permeability has been reported among ASD patients and it is well established that disturbances in eating patterns may cause gut microbiome imbalance (i.e., dysbiosis). Therefore, studies focusing on the potential relationship between gut microbiota and ASD are emerging. We compared the intestinal bacteriome and mycobiome of a cohort of ASD subjects with their non-ASD siblings. Differences between ASD and non-ASD subjects include a significant decrease at the phylum level in *Cyanobacteria* (0.015% vs. 0.074%, *p* < 0.0003), and a significant decrease at the genus level in *Bacteroides* (28.3% vs. 36.8%, *p* < 0.03). Species-level analysis showed a significant decrease in *Faecalibacterium prausnitzii*, *Prevotella copri*, *Bacteroides fragilis*, and *Akkermansia municiphila*. Mycobiome analysis showed an increase in the fungal *Ascomycota* phylum (98.3% vs. 94%, *p* < 0.047) and an increase in *Candida albicans* (27.1% vs. 13.2%, *p* < 0.055). Multivariate analysis showed that organisms from the genus *Delftia* were predictive of an increased odds ratio of ASD, whereas decreases at the phylum level in *Cyanobacteria* and at the genus level in *Azospirillum* were associated with an increased odds ratio of ASD. We screened 24 probiotic organisms to identify strains that could alter the growth patterns of organisms identified as elevated within ASD subject samples. In a preliminary in vivo preclinical test, we challenged wild-type Balb/c mice with *Delftia acidovorans* (increased in ASD subjects) by oral gavage and compared changes in behavioral patterns to sham-treated controls. An in vitro biofilm assay was used to determine the ability of potentially beneficial microorganisms to alter the biofilm-forming patterns of *Delftia acidovorans*, as well as their ability to break down fiber. Downregulation of cyanobacteria (generally beneficial for inflammation and wound healing) combined with an increase in biofilm-forming species such as *D. acidovorans* suggests that ASD-related GI symptoms may result from decreases in beneficial organisms with a concomitant increase in potential pathogens, and that beneficial probiotics can be identified that counteract these changes.

## 1. Introduction

Autism Spectrum Disorder (ASD) is a complex developmental disorder characterized by restricted and repetitive patterns of behaviors or interests resulting in persistent deficits in social communication and interactions [1]. Surveillance to estimate the prevalence of ASD in children conducted by the Autism and Developmental Disabilities Monitoring (ADDM) Network (Centers for Disease Control and Prevention-funded program) in 2016 showed that across 11 sites in the U.S., ASD prevalence was 1 in 54 (18.5 per 1000) children aged 8 years with nearly 4× higher incidence in males [2,3]. In 2020, the reported prevalence was 1 in 36 children aged 8 years [4]. Based on previous data from ADDM, the prevalence of ASD has increased from 1/150 in 2000 to 1/36 in 2020 [3]. Similar findings were reported in other studies [5,6,7,8]. The proposed explanation for this trend was mostly related to factors such as broadening diagnostic criteria, better identification, increased parent awareness, and an increase in other environmental factors such as births to older parents; however, a true increase in the prevalence cannot be ruled out [9].

In addition to pathophysiologic associations with ASD, other major medical disorders such as anxiety, presence of epilepsy [10], immune dysregulation, and gastrointestinal (GI) abnormalities [11,12,13,14,15], as well as other neuropsychiatric [16,17,18] symptoms, are prevalent. It is also notable that GI abnormalities such as gastroesophageal reflux, bloody stools, vomiting, and inflammation with lymphoid nodular hyperplasia have previously been frequently reported to be associated with ASD [11,19,20,21]. A potential contributing factor for GI issues may be the characteristic repetitive behaviors and insistence on sameness of ASD, which results in a limited repertoire of food choices that consequently impacts the patient’s ability to feed effectively, resulting in likely nutritional imbalances [22,23]. Furthermore, de Magistris et al. reported abnormal intestinal permeability among patients with autism (36.7%) and their relatives (21.2%) compared with non-autistic control subjects (4.8%) [24], an observation that has also been reported by D’Eufemia et al. [20]. Since it is well established that disturbances in eating patterns may cause imbalance in the gut microbiome (i.e., dysbiosis) [25], studies focusing on the potential relationship between the gut microbiome and ASD are emerging [26,27,28,29]. Recent studies demonstrated that the gut and brain have a bi-directional interaction through what is now commonly referred to as the gut–brain axis [30], a pathway that has been demonstrated to regulate many neurological diseases such as Multiple Sclerosis, Alzheimer’s Disease, Parkinson’s Disease, schizophrenia, depression, and anxiety, in addition to ASD [31].

Alterations in the abundance of bacteria such as *Prevotella*, *Firmicutes*, *Clostridiales*, and *Bifidobacterium* species were observed in children with ASD compared to non-sibling controls [32]. A murine study by Malkova et al. showed that offspring of immune-activated mothers, wherein the maternal immune system was stimulated by injecting the viral mimic and Toll-like Receptor ligand (TLR) polyinosinic:polycytidylic acid (poly(I:C)) during pregnancy, exhibit core symptoms of ASD [33]. Subsequent studies on the offspring of these studies demonstrated significant deficiency in intestinal barrier integrity, increased gut permeability and abnormal intestinal cytokine profiles, like conditions observed in human ASD subjects [34]. These observations were noticed in 3-week-old maternal immune activation (MIA) offspring, indicating that the abnormalities are established during early life [35]. In addition, transplantation of gut microbiota from human ASD donors into germ-free mice induced autistic behaviors in the form of increased repetitive behavior, decreased locomotion, and decreased communication [36].

Significant improvements in autism severity and gastrointestinal symptoms were reported following the consumption of probiotics, consisting of *Lactobacillus acidophilus*, *L. rhamnosus*, and *Bifidobacteria longum*, in ASD patients [37], results repeated in additional studies [38,39]. However, in another study using probiotics comprising *Lactobacillus plantarum WCSD1*, no significant improvements in behavior were observed in ASD children compared to ASD children not consuming the probiotic [40]. Taken together, these data support the relationship between gut microbiome and ASD symptomology and suggest that additional studies are needed to allow identification of probiotic strains that directly affect the dysbiosis in ASD, which may potentially lead to new therapeutic approaches.

In the present study, we compared the gut bacteriome and mycobiome of a registry of subjects with ASD to those of their non-ASD siblings, with the aim of comparing the human gut microbiome (i.e., bacteriome and mycobiome) to ASD and non-ASD related siblings. We identified significant changes in bacteria (e.g., *Delftia* and *Cyanobacteria*) as well as fungal organisms (e.g., *Candida albicans*) in ASD subjects compared to non-ASD siblings. In a preliminary preclinical test, wild-type Balb/c mice were challenged with *Delftia acidovorans* (increased in ASD subjects) by oral gavage, and changes in behavioral patterns compared to sham-treated controls were evaluated. Based on the microbiota dysbiotic profile in the ASD group, we screened 24 potential probiotic organisms to identify strains that could affect organisms associated with ASD, behavioral and/or gut issues. In vitro assays were used to determine whether selected probiotic strains exhibit the ability to break down fiber and limit biofilm formation from endogenous organisms found to inhabit the gut of ASD subjects. An in vitro biofilm assay was used to determine the ability of potentially beneficial microorganisms to alter the biofilm-forming patterns of *Delftia acidovorans*. The observed downregulation of *Cyanobacteria* combined with an increase in biofilm-forming species (e.g., *D. acidovorans*) suggests that GI symptoms may result from decreased beneficial organisms and a concomitant increase in potential pathogens. Therefore, probiotic supplementation with organisms that can rebalance beneficial microbiota may improve GI symptoms associated with ASD.

## 2. Materials and Methods

### 2.1. Study Cohorts

We analyzed the bacteriome and mycobiome in subjects with self-reported autism and their non-autistic siblings (*n* = 76 subjects), including both local and national volunteers recorded through the Microbiome and Mycobiome Registry of Volunteers with and without Autism Spectrum Disorder at University Hospitals, Cleveland Medical Center (UHCMC IRB #20191290). Potential volunteers were identified through a log of individuals who voluntarily sent stool samples to BIOHM^®^, and who were made aware that they could be contacted for future research if they sent in their samples. Informed consent was distributed along with the questionnaire by email or hardcopy in the form of a written consent document. We requested a waiver of signed consent, so the subjects did not have to send the consent form back. It was denoted that participation in the questionnaire would confirm their consent.

A lifestyle and personal history questionnaire was sent to volunteers after being informed of the study through email, telephone, or mail. The questionnaire was distributed electronically through a REDCap survey link or via a hardcopy version. The questionnaire was used to collect demographic data (i.e., sex, age, race, and ethnicity) of the volunteers as well as other current medical conditions and what medications (e.g., prescribed, OTC, herbal, etc.) they consume. Identifiable information was in the form of name, email address, phone number, and mailing address needed for contact; however, data were de-identified after receipt of the completed questionnaire. Data collected from these volunteers and their stool samples were used to help create a descriptive microorganism DNA profile of their gut microbiome (i.e., bacteriome and mycobiome) composition.

### 2.2. Stool Samples for Microbiome Testing

Multi-kingdom microbiome sequencing consisting of bacteriome and mycobiome was performed on stool samples obtained from children with autism and their non-autistic siblings (*n* = 76 subjects) based on the paper by Hoarau et al. [41]. Sample intake was performed between 21 January 2019 and 19 July 2019. Upon receipt by the lab, stool swab samples were added to 1mL Inhibitex buffer (Qiagen, Germantown, MD, USA) and frozen at −20 °C until all samples had been received. Sample processing and sequencing was performed one month after the last sample was collected.

### 2.3. DNA Extraction

DNA was extracted using a QIA amp Fast DNA Stool Mini kit (Qiagen GmpH, Hilden, Germany) according to the manufacturer’s instructions. Stool swabs collected from subjects and previously frozen in 1 mL of InhibitEX lysis buffer were then thawed at room temperature and incubated for 1 h at 75 °C. Samples were chemi-mechanically disrupted using Fastprep 96 (MP Biomedicals, Solon, OH, USA) in 2 rounds for 300 s at a speed of 1800 RPM. Equal amounts of 100% ethanol and lysate were mixed in a collecting tube and passed through HiBind DNA Mini Columns (Omega Bio-tek, Norcross, GA, USA) with the resulting DNA elution using 80 uL of molecular-grade water. The quality and purity of the isolated genomic DNA were confirmed by gel electrophoresis and quantitated with the Qubit 2.0 instrument applying the Qubit dsDNA HS Assay (Life Technologies, Carlsbad, CA, USA). DNA samples were stored at −20 °C.

### 2.4. Targeted Amplification

Amplifications of the 16S (Region V3–V4) and 5.8S rRNA genes (ITS) were performed using 16S-515 (5′-GGA CTA CCA GGG TAT CTA ATC CTG-3′) and 16S-804 (5′-TCC TAC GGG AGG CAG CAG T-3′) and ITS1 (5′-(TCC GTA GGT GAA CCT GCG G-3′) and ITS4 (5′-TCC TCC GCT TAT TGA TAT GC-3′) primers, respectively. The PCR mixture comprised Q5 High-Fidelity Master Mix (New England Biolabs, Ipswich, MA, USA) with 0.05 μL/mM of each primer; 100 ng of DNA was added to each 50 μL reaction for PCR. Thermo-cycling conditions, using a Biometra Tadvanced thermocycler (Analytik Jena, Jena, Germany), consisted of an initial denaturation step (3 min at 98 °C), followed by 30 cycles of denaturation (10 s at 98 °C), annealing (10 s at 55 °C for the 16S primers and 20 s at 58 °C for the ITS primers), extension (10 s at 72 °C), and a final extension step of 3 min at 72 °C. For validation, the PCR product was separated and visualized qualitatively using gel electrophoresis on 1.5% agarose gel (containing 7 μg/mL ethidium bromide) and screened.

### 2.5. Library Preparation and Sequencing

The amplicon library was cleaned and barcoded followed by emulsion PCR using Ion Torrent S5 Prime workflow according to the manufacturer’s instructions (Thermo Fisher Scientific, Oakwood, OH, USA). Equal volumes of bacterial 16S rRNA and fungal ITS amplicons were pooled, cleaned with AMPure XP beads (Beckman Coulter, Hebron, KY, USA) to remove unused primers, and then exposed to an end repair enzyme for 20 min at room temperature. After an additional AMPure clean up, ligation was performed at 25 °C for 30 min using Ion Torrent P1 and a unique barcoded ‘A’ adaptor. All separate barcoded samples were then pooled in equal amounts (10 μL) and size-selected for the anticipated 16S and ITS range (200–800 bp) using Pippin Prep (Sage Bioscience, Beverly, MA, USA). The library was amplified for seven cycles and quantitated on a StepOne qPCR instrument (Thermo Fisher Scientific, Oakwood, OH, USA) ahead of proper dilution to 100 pM going into an IonSphere templating reaction on the Ion Chef. Library sequencing was completed on an Ion Torrent S5 sequencer (Thermo Fisher Scientific, Oakwood, OH, USA) and barcode-sorted samples were analyzed in our custom pipeline based on Greengenes V13_8 and UNITE database V7.2, designed for the taxonomic classification of 16SrRNA and ITS sequences, respectively. Sequencing reads were clustered into operational taxonomic units (OTUs, 3% distance), described by community metrics, and taxonomically classified within the Qiime 1.8 bioinformatics pipeline.

### 2.6. Statistical Analysis

Statistical analysis was performed using the statistical programming language R (version 3.3.0).

Changes in phylum, genus and species abundance at the community level were assessed using the non-parametric multivariate distance-based analysis of variance using Bray–Curtis (BC) distance for the dissimilarity metric along with its standardized binary form (SBC).

Diversity was analyzed in an unbiased manner using the Shannon diversity index, a measure of abundance considering microbial distribution. Richness was also assessed, reflecting the microbial counts of the bacterial and fungal communities in each sample.

Non-parametric multivariate distance-based associations between bacterial or fungal communities and outcomes were performed using the Adonis function as implemented in the R package vegan version 2.5-2 using the BC dissimilarity distance metric and its SBC based on presence/absence data instead of abundance. Non-parametric Spearman correlation and a Wilcoxon rank-sum test were used for association with continuous outcomes and binary outcomes, respectively.

Longitudinal analysis was performed using all pairwise Multiple Comparison of Mean Ranks as implemented in the PMCMR plus R package version 1.2.0, employing a Kruskal and Wallis test followed by Bonferroni–Dunn post hoc adjustment. *p* < 0.05 was considered statistically significant for all tests after correcting for multiple comparisons. Correction for multiple testing was performed using the Benjamini–Hochberg adjustment method for multiple testing.

### 2.7. Statistical Modeling

A logistic regression model with LASSO (least absolute shrinkage and selection operator) regularization was used for data dimension reduction, microbiome feature selection, and final model construction [16]. Different approaches were used to perform LASSO regularizations as follows: (1) microbiome (phylum, genus and species) alone: regularization of important bacteria and fungi (phylum, genus and species) that were predictive of autism, selected based on univariate analysis; (2) survey data alone: regularization of important demographic, and diet factors that were predictive of autism, selected based on univariate analysis; (3) microbiome (phylum, genus, species) + survey: regularization of important bacteria, fungi (phylum, genus and species), demographic, and diet factors that were predictive of autism, selected based on univariate analysis; and (4) microbiome (all levels) + survey: regularization of important bacteria, fungi (all taxonomic level), demographic, and diet factors that were predictive selected based on univariate analysis. The model employed for the multivariate analysis was Model 3.

All tests were two-sided and a *p*-value ≤ 0.05 was considered statistically significant. The performance concordance index [C-index] was evaluated for each of the proposed models generated.

### 2.8. Potential Probiotic Screening

Following identification of potential contributors to microbiome imbalance in ASD subjects, we embarked on a primary screening program to identify potentially beneficial microorganisms that may be able to rebalance and maintain gut microbiota by affecting these particular organisms. A description of our approach follows.

### 2.9. Microbial Isolates

Appendix A lists different beneficial bacterial and fungal strains (n = 24 strains) tested in our study; all organisms were obtained from commercial sources.

Fiber Fermentation Assay:Media:

Two different growth media were used to evaluate the ability of the microbial strains to break down fibers: GAM Broth (HiMedia Gifu Anaerobic Media, Thermo Fisher Scientific, Oakwood, OH, USA) and Remel^TM^ Andrade’s Broth Base Control (Remel R060102, Thermo Fisher Scientific, Oakwood, OH, USA) w/o Carbohydrate

2.Test Fibers:

Four different formulations of Andrade’s Broth were used:

a. 1% Inulin from chicory (Sigma Aldrich (-I2255) St. Louis, MO, USA)—Reconstituted 1% Inulin (*w*/*v*) in Andrade’s Base Broth w/o Carbohydrate;

b. 1% Agave Inulin (Nuts.com(—p73756952) Cranford, NJ, USA)—Reconstituted 1% Agave Inulin (*w*/*v*) in Andrade’s Base Broth w/o Carbohydrate;

c. 1% Fructooligosaccharide (FOS)(—Orafti 95™, Beneo, Inc., Mannheim, Germany)—Reconstituted 1% FOS (*w*/*v*) in Andrade’s Base Broth w/o Carbohydrate;

d. Control—Andrade’s Base Broth w/o and w/o Carbohydrate.

3.Bacterial Growth

All isolates were grown in GAM for 96 h in an anaerobic environment using AMG gas (5% CO_2_, 5% H_2_ and 90% Nitrogen) at 37 °C.

4.Evaluation of the ability of strains to break down commercially available common fibers

Isolates were diluted to 1 × 10^6^ cells/mL in Andrade’s Base Broth w/o Carbohydrate and 25 µL was used to inoculate the test fiber solutions.

Each fermentable fiber was scored independently by visual inspection and the sum of the score was reported as the Probiotic Fiber Breakdown Score (PFBS), where the largest sum equates to the strains’ ability to ferment 1% Inulin, 1% Agave Inulin, and 1% Fructooligosaccharide (FOS) fiber molecules when challenged as the sole source of carbon. PFBS was based on the following scale:

Yellow color = 0—no fiber fermentation;

Light Pink = 1—inefficient fiber fermentation;

Pink = 2—good fiber fermentation;

Red/Magenta = 3—very efficient fiber fermentation.

5.*Delftia* biofilm growth in the presence and absence of probiotic filtrates

Based on the PFBS, we selected the top 6 fiber-fermenting strains. These strains were then tested for their ability to inhibit biofilm formation by *Delftia acidovorans*, a known biofilm producer [42].

The organisms were grown in GAM pre-reduced broth under strict anaerobic conditions. Isolates were incubated for 24–48 h at 37 °C. After incubation, the supernatant from all strains was filtered through a 0.22 µM filter. Next, the filtrate from each candidate microorganism was combined with GAM broth (1:1) for testing against *Delftia* biofilms.

Sterile 15 mm silicone disks were soaked in fetal bovine serum (FBS) and incubated overnight at 37 °C. *Delftia acidovorans* was grown in GAM pre-reduced media. Using a nephelometer, 1 × 10^7^ cells/mL of *Delftia acidovorans* was suspended in phosphate-buffered saline (PBS). Individual disks were placed in the wells of a 12-well culture plate and 4 mL of *Delftia acidovorans* cell suspension (1 × 10^7^ cells/mL) was added. The disks were then incubated at 37 °C for 90 min. After 90 min, the disks were transferred to single wells in a 24-well plate containing 1.5 mL of the candidate filtrate and GAM broth mix from each candidate probiotic. GAM broth alone was added to a set of disks as positive growth controls. Disks were placed on a rocker and incubated at 37 °C for 96 h. After 96 h, each disk was placed in 2 mL of PBS; the biofilm was removed using a cell scraper and the cells were suspended. Serial dilutions were made and plated for enumeration of colony forming units (CFUs). Each sample was tested in triplicate (technical replicates) against the *Delftia* biofilm. The average log CFUs ± SD for each candidate probiotic strain was compared to the positive growth control; *p*-values of ≤0.05 were considered significant.

6.Experimental Animal Model

Six-week-old female and male WT mice (Balb/C, *n* = 5/group) with a body weight of ~20 g were challenged orally with 10^7^ and 10^8^ CFUs of *Delftia acidovorans* by oral gavage 4 times on days: 0, 2, 4 and 8. Animals were allowed to acclimate for a minimum of 5 days prior to use. Equal numbers of males and females were used for the experiments. Micro-isolator cages (Allentown Inc., Allentown, NJ, USA) with 1/8-inch corn bedding were used to house the mice; animals were provided with cotton nestlets for environmental enrichment (Envigo, Indianapolis, IN, USA). The mice consumed laboratory rodent diet P3000 (Harlan Teklad, Indianapolis, IN, USA) during the experiments. Environmental controls for the animal room were set to maintain a temperature of 16 to 22 °C, a relative humidity of 30–70%, and a 12:12 light/dark cycle. Control groups were challenged with water only. All experiments were conducted in a blinded manner, without prior knowledge of treatments and mouse groups by the experimenter. Mice were randomized to different interventions using a progressive numerical number. The code for each mouse was known only to the animal caretaker and was revealed at the end of the study.

Experimental mice were monitored daily under the supervision of a staff veterinarian. All relevant IACUC recommendations were followed to achieve the highest standards for animal welfare. All veterinary care met NIH and AAALAC standards for humane care for use of laboratory animals. A trained animal technician monitored the animals daily. Any moribund animals were euthanized following the AVMA Panel on Euthanasia guidelines by CO_2_ inhalation followed by cervical dislocation. The School of Medicine at CWRU has several board-certified veterinarians that oversee the day-to-day operations as well as well-trained animal caretakers. All animal studies performed were carried out in the ARC at CWRU in the BRB building under the supervision of a licensed veterinarian. Mice were housed in a sterile barrier caging system in ventilated racks with sterile barrier (filter top) cages (steam-sterilized cage, food and water) with no more than 5 mice/cage. For behavioral experiments, mice were moved to the dedicated Behavioral Core facility within the ARC. Animal husbandry and research personnel wore protective clothing (gown, face mask, gloves, shoe covers, and hair bonnets) and worked with mice on a horizontal HEPA filtered laminar flow clean work bench. Animals were housed in 20′ × 20′ rooms under negative pressure and were monitored for common murine pathogens via a sentinel program on a quarterly basis.

A behavioral test designed to assess features associated with ASD was used to evaluate the mice. Home-cage observations were performed and recorded to determine the general activity level and behavior patterns of the animal in its normal habitat without experimenter interference. This study was a preliminary observational study only; no statistical endpoints were determined.

## 3. Results

The gut microbiome of autistic individuals is distinct from their non-autistic siblings. Principal component analysis (PCA) showed differential clustering for both the bacteriome and mycobiome profiles of autistic individuals compared to their non-ASD healthy siblings (Figure 1A,B).

Analysis of the relative abundance of the bacteriome in ASD subjects compared to their non-ASD siblings, shown in Figure 2, demonstrated a significant decrease in the *Cyanobacteria* (also known as *Cyanophyta*) phylum (0.015% vs. 0.074%, respectively, *p* < 0.0003). Similarly, a significant decrease in the *Bacteroides* genus was also observed in ASD subjects compared to their non-ASD siblings (28.3% vs. 36.8%, respectively, *p* < 0.03). At the species level, a significant decrease in the relative abundance of beneficial bacteria was noted in ASD subjects, including *Faecalibacterium prausnitzii* (a beneficial bacterium reported to exhibit anti-inflammatory effects [43]); *Prevotella copri* (involved in improved glucose tolerance by promoting increased hepatic glycogen storage [44]); *Bacteroides fragilis* (when given as a probiotic, helped to alleviate autism-like symptoms in murine models [45]), (3.7% vs. 6.8%, respectively), and *Akkermansia municiphila* (1.1% vs. 2.3%, respectively).

We also observed a significant increase in bacteria associated with negative health outcomes in autistic subjects compared to their non-autistic siblings. These organisms include *Ruminococcus gnavus*, shown to cause mucin destruction and to produce glucorhamnan, a molecule shown to induce the production of inflammatory cytokines such as TNFα by dendritic cells, with profound effects on human inflammatory responses [46] (1.9% vs. 0.74%, *p* < 0.06); *Parabacteroides distasonis*, a normal gut commensal associated with metabolic benefits, impacting weight and decreasing hyperglycemia and hyperlipidemia, that can also become an opportunistic pathogen under some circumstances [47] (8.7% vs. 6.4%); and *Pseudomonas fragi*, which can entrap potentially harmful bacteria like *E. coli* during biofilm formation, increasing the likelihood of infection [48] (4.1% vs. 1.5%).

Mycobiome analysis of ASD patients compared to their non-ASD siblings, illustrated in Figure 3, demonstrated an increase in the fungal *Ascomycota* phylum (98.3% vs. 94%, respectively, *p* < 0.047). Similarly, the level of the pathogen *Candida albicans* was also significantly elevated in ASD patients vs. non-ASD sibling controls (27.1% vs. 13.2%, respectively, *p* < 0.055). In contrast, the abundance of *Galactomyces geotrichum* (17.4% vs. 29.2%) in ASD subjects vs. non-ASD subjects was decreased (*p* < 0.039) with a similar decrease in abundance for *Pichia fermentans* (4.4% vs. 6.0%) in ASD vs. non-ASD subjects, respectively (*p* = 0.34). Both fungal species are primary fermenters and are considered beneficial commensal fungi [49,50].

### 3.1. Univariate and Multivariate Statistical Analysis Used to Differentiate ASD from Non-ASD Sibling Cohort

Baseline demographics and survey data for the enrolled ASD Cohort are shown in Appendix A. Results of the clinical data survey are also incorporated.

Statistical analysis using the microbiome data (bacteria and fungi at the phylum, genus and species levels) and survey data (demographic, diet, lifestyle, etc.) were then used to detect their effects on predicting autism using a univariate logistic regression model [44]. Appendix A depicts the univariate analysis of the microbiome data representing factors that were significantly associated with ASD. As shown in Appendix A, sex, history of seizure and upper GI disturbances were statistically associated with autism. Several LASSO logistic models were constructed to examine the ability to predict ASD and to identify important bacteria and fungi that were significantly different in the ASD versus control comparison using univariate analyses. The identified bacteria were p__*Cyanobacteria*, s__*Prevotella nigrescens*, g__*Anaerostipes*, g__*Bacteroides*, s__*Coprococcus eutactus*, g__*Leptothrix*, g__*Shewanella*, and g__*Azospirillum*, while the significant fungi included s__*Galactomyces_geotrichum*, p__*Chytridiomycota*, g__*Geotrichum*, and g__*Metarhizium*.

Significant demographic and diet factors that were predictive of ASD, selected using LASSO logistic regression in univariate analysis, were fish consumption, gender and upper GI disturbances. Combining significant factors from both bacteriome (phylum, genus and species) and mycobiome as well as survey data selected using LASSO logistic modeling identified the following at the bacterial level: p__*Cyanobacteria*, s__*Prevotella nigrescens*, g__*Anaerostipes*, g__*Bacteroides*, g__*Leptothrix*, g__*Shewanella*, g__*Delftia*, and g__*Azospirillum*. At the fungal level, only g__*Metarhizium* remained significant; however, fish intake, seizure history, gender and upper gastrointestinal disturbances remained significantly associated. Interestingly, *Metarhizium* has been used as a biopesticide, and thus, identification of this fungus may be a result of food consumption [45].

Finally, we describe a combined model in which microbiome, demographic, diet, and clinical features, identified using LASSO logistic modeling, are incorporated. The performance (concordance index (C-index) was evaluated for each of the proposed models generated. Although each model had a relatively high C-index, Model 3 was the most robust, as shown in Appendix A, and was therefore used in further testing.

Statistical analyses using the microbiome data (bacteria and fungi at the phylum, genus and species levels) and survey data (demographic, diet, lifestyle, etc.) were performed, and results from multivariable logistic modeling using Model 3 are shown in Table 1. Controlling the effects of other factors (gender, seizure history and history of upper GI disturbances), the odds of having ASD were decreased by about 24% per 0.01 percent increase in the phylum *Cyanobacteria* (*p* = 0.013). Similarly, the odds of having ASD were increased 9.99 times per 0.01 percent increase in the genus *Delftia* (*p* = 0.025). One other genus, *Azospirillum*, reached a statistically significance (*p* = 0.026); therefore, the odds of having ASD were decreased by about 81% per 0.01 percent increase in the genus *Azospirillum*. A receiver operating characteristic (ROC) curve analysis shows that the final model (Model 3) has very good diagnostic performance with a C-index of 0.983 and a cutoff value of 0.6447 of the risk score defined above; the sensitivity and specificity for autism diagnosis were 91% and 100%.

Our results demonstrating ASD subjects with a low abundance of *Cyanobacteria* (*p* = 0.013), bacteria that play an important role in fiber breakdown, and an increase in the abundance of g_*Delftia* (*p* = 0.025), a known biofilm producer, suggested that identifying probiotic strains that can break down fiber and inhibit the ability of *Delftia* to form biofilms [42] may ameliorate GI issues encountered in ASD subjects and boost tolerance to fiber-rich diets, leading to better quality of life for those living with ASD. Therefore, we initiated studies to identify probiotic strains that were able to break down fiber and inhibit the ability of *Delftia acidovarans* to produce biofilms. Appendix A lists the various bacterial and fungal strains (*n* = 24 strains) tested in our study of fiber breakdown and biofilm inhibition.

To identify potential probiotics that may enhance fiber breakdown as a potential mechanism to improve ASD subject GI balance, we investigated the potential for candidate probiotic strains to break down fiber.

Table 2 shows the fermentable fiber scores for 1% Inulin, 1% Agave Inulin, and 1% Fructooligosaccharide (FOS) fiber molecules, as well as the total Probiotic Fiber Breakdown Score (PFBS) for the candidate strains. As shown, *Lactobacillus casei* and *Bifidobacterium longum* subsp. *Infantis* demonstrated the greatest ability to ferment these fibers. *Lactobacillus paracasei*, *Lactobacillus delbrueckii* subsp. *bulgaricus and Lactobacillus salivarius* were also highly efficient at breaking down fiber. *Lactobacillus paracasei* and *Lactobacillus delbrueckii* subsp. *bulgaricus* showed a slight reduction in metabolizing 1% Inulin, while *Lactobacillus salivarius* showed a slight reduction in metabolizing FOS. *Bifidobacterium breve* was effective in breaking down Agave Inulin and FOS, although it was not effective in breaking down Inulin. Based on this preliminary data, we selected six strains with a PFBS of 6 or above (indicated with green highlight) to evaluate their ability to inhibit biofilm formation by *Delftia acidovorans*.

### 3.2. Biofilm Formation

Table 3 shows the *p*-values for the growth of *Delftia acidovorans* biofilms in the presence of the supernatants of the candidate probiotics when compared to the untreated growth control. *p*-values of <0.05 were considered significant and no outliers were observed throughout the analysis. As shown, the supernatant from *Lactobacillus casei*, *Bifidobacterium longum* subsp. *Infantis* and *Lactobacillus paracasei* significantly inhibited *Delftia acidovorans* biofilms (*p*-values < 0.02). *Bifidobacterium breve* also significantly inhibited *Delftia acidovorans* biofilms (*p*-values < 0.001).

Based upon inhibition of biofilm formation and favorable Probiotic Fiber Breakdown Scores (PFBSs), we have identified *Lactobacillus casei*, *Bifidobacterium longum* subsp. *Infantis* and *Bifidobacterium breve* as the top candidates for further evaluation using in vivo and extended in vitro assays.

### 3.3. Delftia Influences Mouse Behavioral Changes

Given that the most prominent associated bacterium with increased odds ratios for ASD was *Delftia*, we also performed a preliminary preclinical in vivo experiment whereby Wt Balb/c mice were challenged orally with 10^7^ and 10^8^ CFUs of *Delftia acidovorans* by oral gavage 4 times on days: 0, 2, 4 and 8. Beginning on day 6, the social behavior of the mice was assessed: Mice were observed in their cages for any changes in normal interactions between the mice, such as sniffing, following, and cuddling, as well as any increase in unusual repetitive behaviors like aggression, circling, biting, mounting, jumping, etc. As shown in the comparative movie provided in Appendix A, whilst control mice exhibit normal behavioral patterns, mice gavaged with *Delftia* exhibited repetitive circling and tail biting as well as repetitive grooming behavior that far exceeded normal standards or behaviors compared to the control group.

## 4. Discussion

Numerous studies have postulated a connection between the microbiota–gut–brain axis in ASD [34,36,51,52,53,54,55,56,57,58,59,60,61]. Although these studies have examined this at the bacteriome level, there is a substantial gap in the knowledge surrounding the mycobiome connection. Our previous work in inflammatory bowel disease (IBD) clearly demonstrated that the fungus *Candida tropicalis* (CT) increased significantly in Crohn’s Disease (CD) patients compared to their unaffected relatives (*p* = 0.032), and was positively correlated with the increased abundance of the bacteria *Serratia marcescens* (SM) and *Escherichia coli* (EC) [41].

Based on our CD study, we initiated a larger microbiome/mycobiome prevalence study that currently has >15,000 subjects enrolled. The ASD subjects were collected as a sub-cohort and analyzed as a discovery cohort for ASD–microbiome interactions involving 76 subjects (NCT03819439). We have identified bacterial and fungal communities in ASD subjects that may be involved in imbalances in the gut microbiome of individuals with ASD. We conducted a multivariate analysis and identified significant microbial features that may contribute to this dysbiosis.

Similar observations have been reported in other studies, but results were variable from one study to the other [62,63,64,65].

However, several specific bacterial species were similarly altered across individual studies. For example, a meta-analysis of nine studies, one of which comprised 254 subjects with ASD, found that children with ASD had lower abundance of *Bacteroides* and *Bifidobacterium*, known for their beneficial effects, and a higher abundance of *Faecalibacterium* in their gut compared to non-ASD control subjects. Furthermore, children with ASD had higher abundance of *Lactobacillus* [64]. In contrast, de Angelis et al. reported lower levels of *Faecalibacterium*, which synthesizes short-chain fatty acids (SCFAs) known to have anti-inflammatory properties [66]. Lower levels of *Bifidobacterium* and higher levels of *Bacteroidetes*, *Lactobacillus*, *Clostridium*, *Desulfovibrio*, *Caloramator*, *and Sarcina* in children with ASD compared to non-autistic controls were also reported [67]. Although individual observations may vary, there appears to be a significant association between gut dysbiosis and ASD through what is known as the gut–brain-axis [64] that should not be ignored when considering potential contributory factors to ASD. Recently, Su and colleagues combined multi-kingdom microbiota signatures using metagenomic sequencing to identify a panel of 31 multi-kingdom and functional markers that demonstrated high diagnostic accuracy for ASD (AUC of 0.91). Interestingly, the majority of microbiota markers were decreased in ASD (bacteria, Archaea, viruses), whereas fungal species were not [68].

A unique observation in our study was an increased abundance of *g_Delftia*, a known biofilm-forming organism [69], in children with ASD compared to their non-autistic siblings. *Delftia* has been reported to cause infections, especially in immunocompromised patients such as those with infective endocarditis [70], ocular infections [71], otitis media, peritonitis, urinary tract infections [72], empyema [73], and nosocomial bacteremia, including central venous catheter-related bacteremia [74]. Recently, it was reported that *Delftia* may cause pneumonia with lung cavity formation [74,75]. Moreover, although rare, there have been reports of fatal *Delftia* infections in immunocompetent patients [74], although it is not a common finding in ASD subjects. Interestingly, in a study by Safak et al., *Delftia* was reported to be statistically significantly higher in patients with epilepsy compared to healthy volunteers [76]. Furthermore, a study by McNeill et al. showed that stimulation of monocytes by *Delftia* spp. significantly increased tumor necrosis factor (TNF) production, a cytokine known to contribute significantly to the inflammation and microbiota alterations in IBD patients [77], indicating that *Delftia* exhibits pro-inflammatory activity [78]. Recently, *Delftia* was also reported to increase in a meta-analysis incorporating metagenomic sequencing datasets (10 publicly available raw amplicon datasets) and an internal cohort of ASD subjects to identify ASD-specific microbiome signatures [79].

In contrast to the increase in *Delftia*, we observed a decrease in the abundance of p_*Cyanobacteria*, organisms that possess anti-inflammatory, antioxidant, and wound-healing properties, are a potential source for antibacterial peptides, and, in non-photosynthetic varieties (e.g., *Melainabacteria*), are associated with breaking down fiber [80,81,82,83]. This decrease may be indicative of a limited diet repertoire often associated with ASD patients [84]. This disturbance in the microbiome composition of ASD subjects increases the likelihood for biofilm formation and subsequently persistent infections, which may explain GI symptoms observed in these subjects [85,86,87]. Furthermore, increased permeability of the intestinal epithelium has been reported in ASD patients [24,88] which is the result of disruption of the gut barrier, thus allowing translocation of intestinal microbes and subsequently causing local and systemic inflammation [89]. A concomitant increase in cytokines, including interleukin-1β (IL-1β), IL-6, interferon-γ, and TNF-α, which in turn cross the blood–brain barrier and cause immune responses in the brain, was also noted [52].

Based on our preliminary data, we tested the activity of 24 bacterial strains to break down fiber using in vitro assays. We then chose six strains that showed the highest activity and tested their ability to inhibit *Delftia* biofilm formation. Of these strains, *Lactobacillus casei*, *Bifidobacterium longub* subps. *Infantis*, and *Bifidobacterium breve* were the most effective in inhibiting the growth of *Delftia* biofilms while being efficient in breaking down fibers. These strains are known for their ability to break down biofilms, as well as inhibiting intestinal infections (such as rotavirus infection), balancing the immune system, improving intestinal barrier function, and reducing gut dysbiosis, as demonstrated in several studies [90,91,92,93,94,95,96,97,98,99,100].

*B. breve* has shown antibacterial activity against *Clostridioides difficile*, a common cause of antibiotic-associated diarrhea and one of the species that were shown to be increased in ASD subjects and play a role in ASD symptomatology, likely induced by the production of metabolic products that are potentially toxic to humans, such as phenols [101]. Additionally, *B. infantis* activity was shown to not be restricted to the mucosal immune system but extended to the systemic immune system by causing marked decreases in pro-inflammatory cytokines in conditions such as ulcerative colitis (UC), chronic fatigue syndrome, and psoriasis [26,102].

Finally, in a preliminary in vivo preclinical test, wild-type Balb/c mice were challenged with 10^7^ and 10^8^ CFUs of *Delftia acidovorans* by oral gavage four times on days: 0, 2, 4 and 8. We observed changes in behavioral patterns in mice receiving *D. acidovorans* compared to sham gavaged controls. Although this experiment used a small number of animals, the behavioral changes were notable and we therefore felt compelled to share these observations, despite the very preliminary nature of the experiments. Obviously, these experiments will require a more robust number of replicates and more controlled behavioral assessments in future studies.

## 5. Conclusions

We identified significantly different abundances of bacterial and fungal microbiota in ASD individuals compared to their non-autistic siblings, suggesting potential dysregulation of the GI microbiome in ASD. The downregulation of *Cyanobacteria* (normally beneficial for inflammation and wound healing) combined with an increase in biofilm-forming species such as *Delftia* suggests that GI symptoms may result from decreases in beneficial organisms with a concomitant increase in potential pathogens. Screening potential probiotics for their ability to alter fiber breakdown and biofilm formation demonstrated a proof of concept that probiotic supplementation with organisms that can rebalance beneficial microbiota may improve GI symptoms by rebalancing the microbiota distribution. A preliminary murine study also demonstrated that behavioral changes could be elicited following delivery of bacteria (*D. acidovorans*) found to be elevated in ASD subjects. Taken together, our findings support additional studies including clinical studies designed to assess a probiotic formulation that may reduce GI symptoms encountered in ASD subjects, and the conduct of human proof-of-concept studies for using probiotics as medical food.

## Figures and Tables

**Figure 1 nutrients-16-03004-f001:**
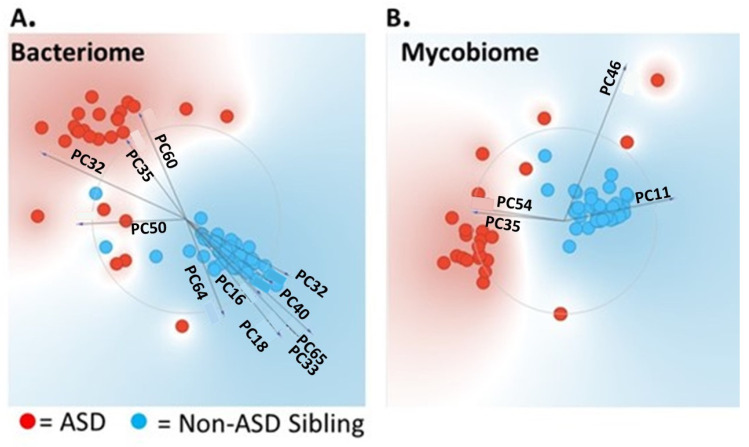
Principal component analysis (PCA) showed differential clustering for both the bacteriome (**A**) and mycobiome (**B**) profiles of autistic individuals compared to their non-ASD healthy siblings.

**Figure 2 nutrients-16-03004-f002:**
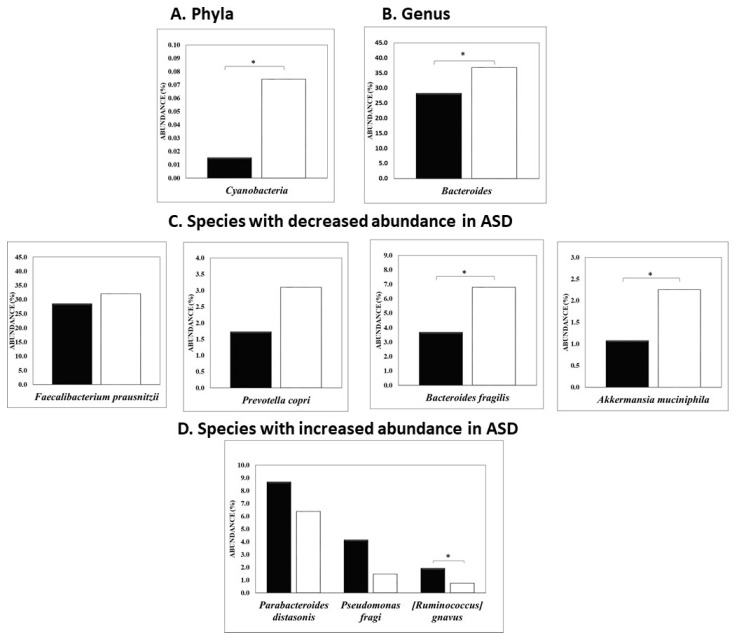
Relative abundance (%) of bacterial phyla, genera, and species associated with ASD (black bars) vs. non-ASD (white bars) siblings. * *p* < 0.05.

**Figure 3 nutrients-16-03004-f003:**
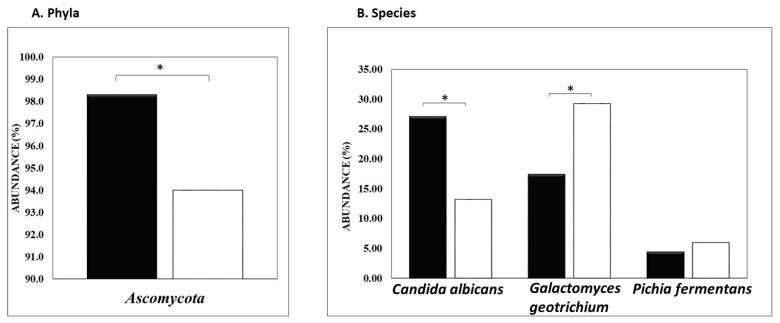
Relative abundance (%) of fungal phylum and species level associated with ASD (black bars) subjects and non-ASD siblings (white bars). * *p* < 0.05.

**Table 1 nutrients-16-03004-t001:** Multivariable logistic regression analysis.

Factors	Multivariable Model	
	Odds Ratio (95% CI)	*p* Value
p__*Cyanobacteria* (per 0.01 percentage of abundance increase)	0.76 (0.62, 0.94)	0.013
s__*Prevotella nigrescens* (per 0.01 percentage of abundance increase)	0.84 (0.64, 1.1)	0.202
g__*Anaerostipes* (per percentage of abundance increase)	0.17 (0.01, 2.53)	0.197
g__*Bacteroides* (per percentage of abundance increase)	0.98 (0.93, 1.03)	0.355
g__*Leptothrix* (per 0.01 percentage of abundance increase)	1.26 (0.8, 1.98)	0.321
g__*Delftia* (per 0.01 percentage of abundance increase)	9.99 (1.33, 75)	0.025
g__*Shewanella* (per 0.1 percentage of abundance increase)	0.37 (0.03, 4.6)	0.443
g__*Azospirillum* (per 0.01 percentage of abundance increase)	0.19 (0.05, 0.82)	0.026
g__*Metarhizium* (per percentage of abundance increase)	0.21 (0.01, 7.25)	0.386
Fish (per serving increase)	0.29 (0.1, 0.82)	0.02
Seizure (Yes vs. no)	1.33 (0.13, 13.78)	0.809
Sex (Male vs. Female)	36.79 (2.89, 467.93)	0.006
upper gastrointestinal disturbances	2.89 (0.25, 32.87)	0.392

**Table 2 nutrients-16-03004-t002:** Fiber fermentation scores of probiotic candidate strains.

Probiotic Candidates	Probiotic Fiber Breakdown Score (PFBS)	Inulin	Agave Inulin	FOS	Growth Rate
*Lactobacillus casei*	9	3	3	3	Good
*Bifidobacterium longum* subps. *Infantis*	9	3	3	3	Good
*Lactobacillus paracasei*	8	2	3	3	Good
*Lactobacillus salivarius*	8	3	3	2	Good
*Lactobacillus delbrueckii* subsp. *bulgaricus*	8	2	3	3	Poor
*Bifidobacterium breve*	6	0	3	3	Good
*Bifidobacterium bifidum*	5	0	2	3	Good
*Pediococcus pentosaceus*	5	0	3	2	Good
*Streptococcus thermophilus*	4	0	3	1	Good
*Pediococcus acidilactici*	4	0	3	1	Good
*Lactobacillus plantarum*	4	0	2	2	Good
*Bifidobacterium longum*	3	0	2	1	Good
*Lactobacillus plantarum*	3	0	2	1	Good
*Lactobacillus rhamnosus*	3	0	2	1	Good
*Lactobacillus rhamnosus*	3	0	2	1	Good
*Bifidobacterium breve*	3	0	2	1	Good
*Saccharomyces boulardii*	3	0	0	3	Good
*Lactobacillus gasseri*	3	0	2	1	Good
*Lactococcus lactis*	1	0	1	0	Good
*Lactobacillus reuteri*	1	0	1	0	Good
*Lactobacillus acidophilus*	1	0	0	1	Good
*Bifidobacterium longum*	1	0	1	0	Good
*Bifidobacterium pseudocatenulatum*	0	0	0	0	Good
*Bifidobacterium catenulatum* subsp. *kashiwanohense*	0	0	0	0	Good
*Komagataella pastoris*	0	0	0	0	Good

**Table 3 nutrients-16-03004-t003:** Growth of *Delftia acidovorans* biofilms in the presence of candidate probiotic supernatants.

Delftia Biofilm Inhibition by Various Organisms	Log CFUs of Delftia Biofilm (±SD)	*p*-Value Compared to the Growth Control
*Untreated Delftia Control*	6.28 ± 0.02	
*Lactobacillus casei*	5.17 ± 0.12	0.002
*Bifidobacterium longum* subps. *infantis*	5.09 ± 0.34	0.025
*Lactobacillus paracasei*	5.79 ± 0.13	0.017
*Lactobacillus salivarius*	6.21 ± 0.04	0.057
*Lactobacillus delbrueckii* subsp. *bulgaricus*	6.27 ± 0.03	0.608
*Bifidobacterium breve*	4.75 ± 0.02	<0.001

## Data Availability

All data generated or analyzed during this study are included in this published article (and its Appendix A). The datasets generated during and/or analyzed during the current study are available from the corresponding author on reasonable request.

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
