# Peer review of "Analysis of Gut Bacterial and Fungal Microbiota in Children with Autism Spectrum Disorder and Their Non-Autistic Siblings"

_nutrients, 2024, doi:10.3390/nu16173004_

Round 1

Reviewer 1 Report

Comments and Suggestions for Authors

The Title “Analysis of Gut Bacterial and Fungal Microbiota in children with Autism Spectrum Disorder and their non-autistic siblings” by

 Mauricio Retuerto et al, offers intriguing and possibly significant data about the gut microbiota and Autism Spectrum Disorder (ASD), with a special emphasis on the function of bacteria and fungus. The study investigates the relationship between distinct microbial communities and ASD, suggests prospective probiotic candidates for additional exploration, and conducts preliminary in vivo trials to examine behavioral changes associated with the microbiome.

The manuscript should be considered by following the few points.

Note: There are more than double space in many sentences, please try to fix it.

1. Abstract: A comparison of the gut bacterial and fungal microbiota of children with ASD and their non-ASD siblings is promised in the title, yet the abstract just briefly describes a few parts of the study. This gap might be filled with a more thorough examination of the fungal microbiome and the findings of the sibling comparison.

2. introduction: Fungal microbiota and sibling comparison are not given enough emphasis at the beginning, which is a little chaotic and introduces ideas like probiotics and mouse models.

3. Sample Collection: The collecting procedure of the stool samples, including the technique, storage conditions, and interval between collection and processing, is not included in this section. This is important since these variables have an impact on the composition of the microbiome.

4. Thermal Cycler Model: Since thermal cyclers might differ somewhat in performance, it can be crucial to know the exact model or manufacturer of the thermal cycler being utilized.

5. Biofilm Assay Reproducibility: It is not stated how the biofilm inhibition assay was made reproducible, e.g., by using technical duplicates or many independent investigations.

6. Details of the Multivariable Logistic Regression Model: There is a dearth of information in the description of the Multivariable Logistic Regression Analysis regarding the model's construction, such as whether interaction terms were taken into account, the method used to determine multicollinearity, and the process for choosing which variables to include in the final model.

7. Standard deviations are included in the data shown in Table 3, but more thorough explanations of the variability across repetitions would be beneficial. Were any outliers found, and if so, how were they treated throughout the analysis?

8. Has the possibility of probiotic supplementation caused any negative effects or dangers been examined, especially for susceptible groups such as those with ASD? In further research, how will they be tracked?

Note: There are more than double space in many sentences, try to fix it.

Reviewer 2 Report

Comments and Suggestions for Authors

The manuscript titled "Analysis of Gut Bacterial and Fungal Microbiota in Children..." is a fairly good piece of work that, with a few minor revisions. Below are my comments:

  1. Please rewrite the first sentence in the abstract, as it sounds a bit awkward.
  2. The last sentence in the abstract also needs revision.
  3. Is the keyword "microbiome" necessary?
  4. "However, in another study using probiotics comprised of Lactobacillus..." - why mention this if there was no improvement?
  5. I also think the introduction should be slightly revised, as the research topic is very broad, and the introduction presented by the authors is rather sparse.
  6. Figure 1 - the PC axis labels are barely visible. Please enlarge them slightly.
  7. Figure 2 - the labels on the Y and X axes are also hard to see; please improve them.
  8. "Moreover, our data also demonstrated differences in the microbiome composition..." - what is the purpose of this discussion?
  9. "In contrast to the increase in Delftia, we observed a decrease in the abundance of p_Cy..." - what could have caused this? It might be worth explaining with a single sentence.
  10. Conclusions - need to be entirely rewritten. Please make it more polished; currently, it seems too vague.

Overall, the work is very good. My recommendation is minor revisions.

Round 2

Reviewer 1 Report

Comments and Suggestions for Authors

I am pleased to inform you that the current version of your work, "Analysis of Gut Bacterial and Fungal Microbiota in Children with Autism Spectrum Disorder and their Non-Autistic Siblings," can be accepted. We value the thorough work you did in responding to all of the reviewers' comments and suggestions. Your careful and insightful revisions have greatly improved the manuscript's quality.

I appreciate your important contribution to the field. We hope to work with you  in the future and are excited to see your work published.